# Robust and accelerated single-spike spiking neural network training with applicability to challenging temporal tasks

## Abstract

Spiking neural networks (SNNs), particularly the single-spike variant in which neurons spike at most once, are considerably more energy efficient than standard artificial neural networks (ANNs). However, single-spike SSNs are difficult to train due to their dynamic and non-differentiable nature, where current solutions are either slow or suffer from training instabilities. These networks have also been critiqued for their limited computational applicability such as being unsuitable for time-series datasets. We propose a new model for training single-spike SNNs which mitigates the aforementioned training issues and obtains competitive results across various image and neuromorphic datasets, with up to a 13.98× training speedup and up to an 81% reduction in spikes compared to the multi-spike SNN. Notably, our model performs on par with multi-spike SNNs in challenging tasks involving neuromorphic time-series datasets, demonstrating a broader computational role for single-spike SNNs than previously believed.

## 1 Introduction

Artificial neural networks (ANNs) have achieved impressive feats over recent years, obtaining human-level performance on visual and auditory tasks (Hinton et al., 2012; He et al., 2016), natural language processing (Brown et al., 2020) and challenging games (Mnih et al., 2015; Silver et al., 2017; Vinyals et al., 2019). However, as the difficulty and complexity of the tasks increase, so has the size of the networks required to solve them, demanding a substantial and unsustainable amount of energy (Strubell et al., 2019; Schwartz et al., 2020). Inspired by the extreme energy efficiency of the brain (Sokoloff, 1960), spiking neural networks (SNNs) emulated on neuromorphic computers attempt to solve this dilemma, requiring significantly less energy than ANNs (Wunderlich et al., 2019). These networks are of growing interest, obtaining noteworthy results on visual (Fang et al., 2021; Zhou & Li, 2021), auditory (Yin et al., 2020; Yao et al., 2021) and reinforcement learning problems (Patel et al., 2019; Tang et al., 2020; Bellec et al., 2020).

A particular class of SNNs in which individual neurons respond with at most one spike aims to further amplify the energy and scaling advantages of standard SNNs and ANNs. Inspired by the sparse spike processing shown to exist at least for certain stimuli in the auditory and visual systems (Heil, 2004; Gollisch & Meister, 2008), and forming a class of universal function approximator (Comsa et al., 2020), these networks obtain extreme energy efficiency due to their single-spike nature (Oh et al., 2021; Liang et al., 2021). Although providing a promising path toward building very large and energy-efficient networks, we are yet to understand how to properly train these SNNs. The success of the backprop training algorithm in ANNs does not naturally transfer to single- and multi-spike SNNs due to their non-differentiable activation function. Current attempts at training are either slow (as time is sequentially simulated) or suffer from training instabilities (e.g. the dead neuron problem) and idiosyncrasies (e.g. requiring particular regularisation) (Eshraghian et al., 2021). Additionally, it has been argued that single-spike networks have limited applicability and are not suited for temporal problems, as recently pointed out by Eshraghian et al. (2021): "[...] it enforces stringent priors upon the network (e.g., each neuron must fire only once) that are incompatible with dynamically changing input data" and Zenke

et al. (2021): "[...] only using single spikes in each neuron has its limits and is less suitable for processing temporal stimuli, such as electroencephalogram (EEG) signals, speech, or videos".

In this work we address these shortcomings by proposing a new model for training single-spike networks, for which the main contributions are summarised as follows.

1. Our model for training single-spike SNNs eschews all sequential dependence on time and exclusively relies on GPU parallelisable non-sequential operations. We experimentally validate this to obtain faster training times over sequentially trained control models on synthetic benchmarks (up to 16.77× speedup) and real datasets (up to 13.98× speedup).

2. We obtain competitive accuracies on various image and neuromorphic datasets with extreme spike sparsity (up to 81% fewer spikes than standard multi-spike SNNs), with our model being insensitive to the dead neuron problem and not requiring careful network regularisation. In other single-spike training methods, but not in our model, the dead neuron problem tends to halt learning due to reduced network activity.

3. We showcase our model's applicability in deeper and convolutional networks, and through the inclusion of trainable membrane time constants manage to solve difficult temporal problems otherwise thought to be unsolvable by single-spike networks.

## 2 BACKGROUND AND RELATED WORK

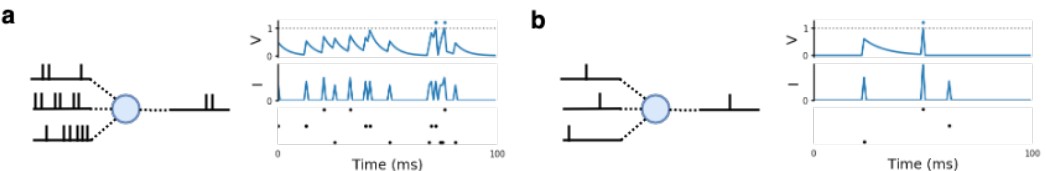

Figure 1: Spiking neuron dynamics. **a.** Left: A multi-spike neuron emitting and receiving (per presynaptic terminal) multiple spikes. Right: Input and output activity of the neuron (bottom panel: Input raster, middle panel: Input current $I$ and top panel: Membrane potential $V$. Dotted line represents the firing threshold and a dot above denotes a spike). **b.** Left: A single-spike neuron emitting and receiving (per presynaptic terminal) at most one spike per stimulus. Right: Input and output activity of the neuron).

### 2.1 SINGLE-SPIKE MODEL

A spiking neural network (SNN) consists of artificial neurons which output binary signals known as spikes (Figure 1a). Assume a feedforward network architecture of $L$ fully connected layers, where each layer $l$ consists of $N^{(l)}$ neurons that are fully connected to the next layer $l+1$ via synaptic weights $W^{(l+1)} \in \mathbb{R}^{N^{(l+1)} \times N^{(l)}}$. Neuron $i$ in layer $l$ emits a spike $S_i^{(l)}[t] \in \{0, 1\}$ at time $t$ if its membrane potential $V_i^{(l)}[t] \in \mathbb{R}$ reaches firing threshold $V_{th}$.

$$S_i^{(l)}[t] = f(V_i^{(l)}[t]) = \begin{cases} 1, & \text{if } V_i^{(l)}[t] > V_{th} \\ 0, & \text{otherwise} \end{cases} \tag{1}$$

Membrane potentials evolve according to the leaky integrate and fire (LIF) model

$$\tau \frac{dV_i^{(l)}(t)}{dt} = -V_i^{(l)}(t) + V_{rest} + RI_i^{(l)}(t) \tag{2}$$

where $\tau \in \mathbb{R}$ is the membrane time constant and $R \in \mathbb{R}$ is the input resistance (Gerstner et al., 2014).[1] Without loss of generality the LIF model is normalised ($V_i^{(l)}(t) \in [0, 1]$ by $V_{rest} = 0, V_{th} = 1, R = 1$; see Appendix) and discretised using the forward Euler method (see Appendix), from

---

[1]Note, we use () to refer to continuous time and [] to refer to discrete time.

which the membrane potential can be computed at every discrete simulation time step $t \in \{1, \ldots, T\}$ for $T \in \mathbb{N}$ using the difference equation below.

$$V_i^{(l)}[t+1] = \beta V_i^{(l)}[t] + (1-\beta) \underbrace{\left( b_i^{(l)} + \sum_{j=1}^{N^{(l-1)}} W_{ij}^{(l)} S_j^{(l-1)}[t+1] \right)}_{\text{Input current } I_i^{(l)}[t+1]} - \underbrace{S_i^{(l)}[t]}_{\text{Spike reset}} \qquad (3)$$

The membrane potential is charged from the current induced by the incoming presynaptic spikes $S^{(l-1)}[t] \in \mathbb{R}^{N^{(l-1)}}$ and from the constant bias current source $b_i^{(l)}$. Over time, this potential dissipates, and the degree of dissipation is captured by $0 \le \beta = \exp(\frac{-\Delta t}{\tau}) \le 1$ (for simulation time-step size $\Delta t \in \mathbb{R}$). The neuron's membrane potential is at resting state $V_{rest} = 0$ in the absence of any input current and emits a spike $S_i^{(l)}[t] = 1$ if the potential rises above firing threshold $V_{th} = 1$ (after which it is reduced back close to resting state).

To enforce the single-spike constraint, we keep track if a neuron has spiked prior to time $t$ using the variable $d_i^{(l)}[t] = \max(S_i^{(l)}[t-1], d_i^{(l)}[t-1])$, which is zero before the first spike and one thereafter ($d_i^{(l)}[t=0] = 0$). We then redefine the output spikes as $\tilde{S}_i^{(l)}[t] = (1 - d_i^{(l)}[t]) \cdot S_i^{(l)}[t]$, thus ensuring that no more than a single spike is emitted during simulation (Figure 1b).

## 2.2 Single-spike training techniques

The main problem with training single- and multi-spike SNNs is the non-differentiable nature of their activation function. This precludes the direct use of the backprop algorithm (Rumelhart et al., 1986), which has underpinned the successful training of ANNs. Various SNN training solutions have been proposed, which we group into three categories.

**Shadow training**   Instead of directly training a SNN, an already trained ANN is mapped to a SNN. This approach has actively been explored in the multi-spike setting (O'Connor et al., 2013; Esser et al., 2015; Rueckauer et al., 2016; 2017), with recent work extending this to single-spike networks (Stöckl & Maass, 2019; Park et al., 2020). Although these approaches permit the training of large networks, they come with various shortcomings. Some shortcomings are method specific, such as Stöckl & Maass (2019) who outline how a single ANN unit can be represented as a network of spiking units. However, this leads to an undesirable blowup of network parameters in their conversion process (which is avoided by our approach). Other shortcomings are more general, such as the lack of support for training neural parameters besides synaptic weights (which our approach permits) or inference accuracy being lost in the conversion process, where mapped SNNs perform worse than the original ANNs (which we avoid).

**Training using the spike times**   An approach used to directly train SNNs using backprop involves passing gradients through the time of spiking, which sidesteps the aforementioned non-differentiability issue (Bohte et al., 2002; Mostafa, 2017; Comsa et al., 2020; Kheradpisheh & Masquelier, 2020; Zhang et al., 2021; Zhou & Li, 2021; Zhou et al., 2021). Although commonly used for training single-spike SNNs, this approach suffers from various shortcomings, such as 1. the dead neuron problem, where a lack of spiking activity halts the learning process (which we overcome), 2. being usually constrained to integrate and fire (IF) neurons (where we support both the IF and LIF model), 3. having performance dependent on the computationally costly processing of presynaptic spikes using postsynaptic potential (PSP) kernels (which we show not to be necessary) and 4. requiring careful network regularisation (which we avoid).

**Training using the membrane potentials**   Another approach to directly training SNNs using backprop is by replacing the undefined gradient of the non-differentiable spike function with a surrogate gradient (Esser et al., 2016; Hunsberger & Eliasmith, 2015; Zenke & Ganguli, 2018; Lee et al., 2016), which permits the flow of gradient through every membrane potential in time (Bellec et al., 2018; Shrestha & Orchard, 2018; Neftci et al., 2019). This method has been shown to circumvent the dead neuron problem and permit the training of other neural parameters besides synaptic connectivity (such as membrane time constants) that have been shown to improve network performance (Perez-Nieves et al., 2021). However, these results have not been replicated in

the single-spike setting (which we do). A shortcoming of this method is its slow training speed, as the network needs to sequentially be simulated at every point in time (which we overcome).

# 3 PROPOSED TRAINING SPEEDUP ALGORITHM

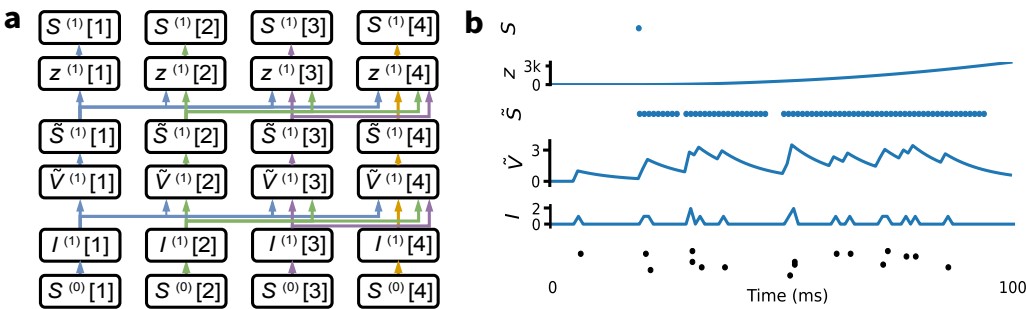

Figure 2: Illustration of our model. **a.** The computational graph of our model for 4 time steps. Input spikes $\mathbf{S}^{(0)}$ induce currents $\mathbf{I}^{(1)}$, which charge the membrane potential without reset $\tilde{\mathbf{V}}^{(1)}$. These no-reset membrane potentials are mapped to erroneous output spikes $\tilde{\mathbf{S}}^{(1)}$, which are then transformed to a latent representation $\mathbf{z}^{(1)}$ encoding an ordering of spikes and finally mapped to the correct output spikes $\mathbf{S}^{(1)}$ (same coloured edges denote output from same source). **b.** Example activity of our model throughout the different stacks of processing.

We propose a new model for training SNNs in which individual neurons spike at most once. Our solution overcomes the slow training speeds of prior training algorithms by eschewing all sequential dependence and recasting the standard single-spike model to exclusively rely on non-sequential operations. Although our model performs more calculations than the standard single-spike model, all these calculations are highly parallelisable and thus substantially faster to train (see Appendix). Our model is comprised of three main steps which are readily implementable in modern auto differentiation frameworks (Abadi et al., 2016; Paszke et al., 2017; Bradbury et al., 2018). For illustration purposes, we provide a diagram of the model's computational graph (Figure 2a) and an example of how input spikes are transformed throughout the model's different layers of processing (Figure 2b).

**1. Convert presynaptic spikes to input current** As in the standard model, we map the time series of presynaptic spikes $\mathbf{S}^{(l-1)}_j$ [2] to a time series of input currents $\mathbf{I}^{(l)}_i$, which is achieved using a tensor multiplication.

$$I^{(l)}_i[t] = \sum_{j=1}^{N^{(l-1)}} W^{(l)}_{ij} S^{(l-1)}_j[t] \tag{4}$$

**2. Calculate membrane potentials without reset** In contrast to the standard model, we calculate modified membrane potentials $\tilde{\mathbf{V}}^{(l)}_i$ from the input current $\mathbf{I}^{(l)}_i$ by excluding the reset mechanism. By dropping the reset term $-S^{(l)}_i[t]$ in Equation 3 and unrolling this altered equation (see Appendix), we obtain a convolutional form allowing us to calculate these no-reset membrane potentials $\tilde{\mathbf{V}}^{(l)}_i$ without any sequential operations (where $\boldsymbol{\beta} = [\beta^0, \beta^1, \cdots, \beta^{T-1}]$).

$$\tilde{V}^{(l)}_i[t] = \beta^t V^{(l)}_i[0] + (1-\beta) \sum_{k=1}^{t} \beta^{t-k} I^{(l)}_i[k] \tag{5}$$
$$= \beta^t V^{(l)}_i[0] + (1-\beta)\big(\mathbf{I}^{(l)}_i \circledast \boldsymbol{\beta}\big)[t]$$

**3. Map no-reset membrane potentials to output spikes** We map the time series of no-reset membrane potentials $\tilde{\mathbf{V}}^{(l)}_i$ to output spikes $\mathbf{S}^{(l)}_i$ (which contains at most one spike). We obtain a

---

[2]Bold face variables denotes arrays as opposed to scalar values.

time series of erroneous output spikes $\tilde{\mathbf{S}}_i^{(l)}$ by passing no-reset membrane potentials $\tilde{\mathbf{V}}_i^{(l)}$ through the spike function $f$ (Equation 1)

$$\tilde{S}_i^{(l)}[t] = f(\tilde{V}_i^{(l)}[t]) \tag{6}$$

Due to the removal of the spike reset mechanism, only the first spike occurrence in $\tilde{\mathbf{S}}_i^{(l)}$ follows the dynamics set out by the LIF model (Equation 3) and thus all spikes succeeding the first spike occurrence are removed (compliant with the single spike assumption). We achieve this by constructing correct output spikes $\mathbf{S}_i^{(l)}$ with $S_i^{(l)}[t] = 0$ for $t \in \{1, 2, \ldots, T\}$ except $S_i^{(l)}[t] = 1$ for the smallest $t$ satisfying $\tilde{V}_i^{(l)}[t] > 1$ (if such $\tilde{V}_i^{(l)}[t]$ exists). A straightforward solution would be to iterate over all elements in $\tilde{\mathbf{S}}_i^{(l)}$ and set all spikes succeeding the first to zero, but such sequential calculation is the very problem we set out to remediate. We propose a vectorised solution to this problem which is comprised of two steps:

1. Map the erroneous output spikes $\tilde{\mathbf{S}}_i^{(l)}$ to a latent representation $\mathbf{z}_i^{(l)} = \phi(\tilde{\mathbf{S}}_i^{(l)})$, where every element therein encodes an ordering of the spikes. This is achieved by passing the erroneous output spikes $\tilde{\mathbf{S}}_i^{(l)}$ through proposed function $\phi$ (Proposition 1), which maps all elements besides the first spike occurrence to a value other than one ($z_i^{(l)}[t] \neq 1$ for all $t$ except for the smallest $t$ satisfying $\tilde{S}_i^{(l)}[t] = 1$ if such $t$ exists).

2. Obtain the correct output spikes $\mathbf{S}_i^{(l)} = g(\mathbf{z}_i^{(l)})$ by passing the latent representation $\mathbf{z}_i^{(l)}$ through function $g$, which uses the encoded spike ordering to produce the correct outputs spikes $\mathbf{S}_i^{(l)}$ by mapping every value besides one to zero. [3]

$$g(\mathbf{z}_i^{(l)})[t] = \begin{cases} 1, & \text{if } z_i^{(l)}[t] = 1 \\ 0, & \text{otherwise} \end{cases} \tag{7}$$

**Proposition 1.** *Function* $\phi(\tilde{\mathbf{S}}_i^{(l)})[t] = \sum_{k=1}^t \tilde{S}_i^{(l)}[k](t-k+1)$ *acting on* $\tilde{\mathbf{S}}_i^{(l)} \in \{0, 1\}^T$ *contains at most one element equal to one* $\phi(\tilde{\mathbf{S}}_i^{(l)})[t] = 1$ *for the smallest t satisfying* $\tilde{S}_i^{(l)}[t] = 1$ *(if such t exists).*

*Proof.* Firstly, if $\tilde{S}_i^{(l)}[t] = 0$ for all $t \in [1, T]$ then $\phi(\tilde{\mathbf{S}}_i^{(l)})[t] = 0$ for all $t \in [1, T]$ (follows from substitution). Secondly, if $\tilde{S}_i^{(l)}[t_1] = 1$ for smallest $t_1 \in [1, T]$ then $\phi(\tilde{\mathbf{S}}_i^{(l)})[t_1] = 1$ (follows from substitution) and there can exist no $t_2 > t_1$ such that $\phi(\tilde{\mathbf{S}}_i^{(l)})[t_2] = 1$ as

$$\begin{aligned} \phi(\tilde{\mathbf{S}}_i^{(l)})[t+1] &= \sum_{k=1}^{t+1} \tilde{S}_i^{(l)}[k]\big((t+1)-k+1\big) \\ &= \sum_{k=1}^{t} \tilde{S}_i^{(l)}[k]\big((t+1)-k+1\big) + \tilde{S}_i^{(l)}[t+1] \\ &= \sum_{k=1}^{t} \tilde{S}_i^{(l)}[k](t-k+1) + \sum_{k=1}^{t} \tilde{S}_i^{(l)}[k] + \tilde{S}_i^{(l)}[t+1] \\ &= \phi(\tilde{\mathbf{S}}_i^{(l)})[t] + \sum_{k=1}^{t+1} \tilde{S}_i^{(l)}[k] \end{aligned} \tag{8}$$

Thus $\phi(\tilde{\mathbf{S}}_i^{(l)})[t_2] > \phi(\tilde{\mathbf{S}}_i^{(l)})[t_1]$ for all $t_2 > t_1$ as $\sum_{k=1}^{t_2} \tilde{S}_i^{(l)}[k] \geq \sum_{k=1}^{t_1} \tilde{S}_i^{(l)}[k] = 1 > 0$. $\square$

## 4 EXPERIMENTS AND RESULTS

We investigate our model's speedup advantages and performance on real datasets in comparison to prior work. All models were implemented using PyTorch (Paszke et al., 2017) with benchmarks and training conducted on a cluster of NVIDIA Tesla A100 GPUs.

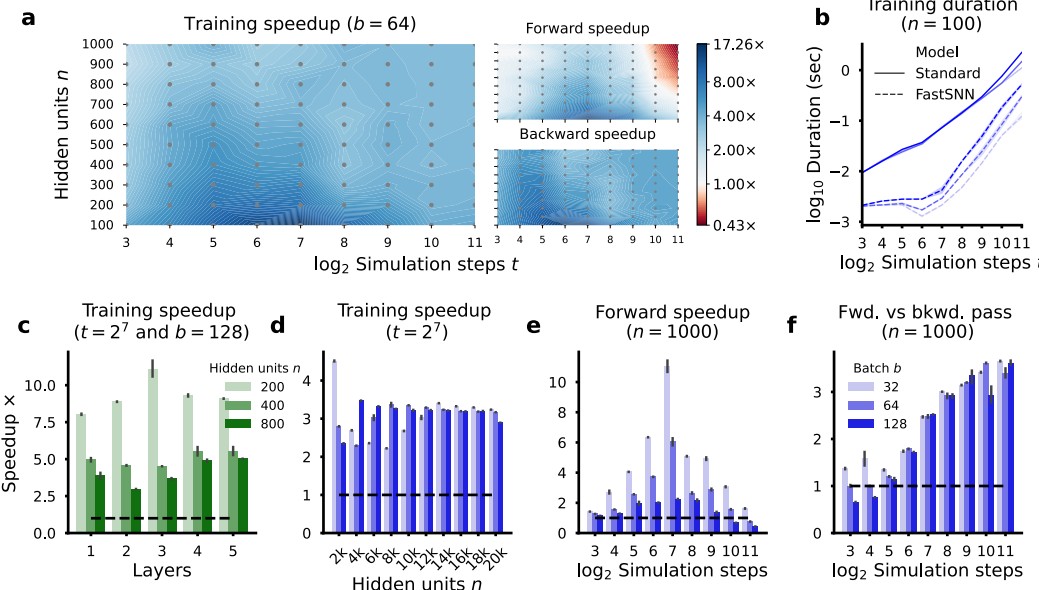

Figure 3: Training speedup of our model over the standard model. **a.** Total training speedup as a function of the number of hidden neurons $n$ and simulation steps $t$ (left), alongside the corresponding forward and backward pass speedups (right). **b.** Training durations of both models for fixed hidden neurons $n = 100$ and variable batch size $b$. **c.** Training speedup over different number of layers for fixed time steps $t = 2^7$ and batch size $b = 128$. **d.** Training speedup over large number of hidden neurons $n$ for fixed time steps $t = 2^7$ and variable batch size $b$. **e.** Forward pass speedup for fixed time steps $t = 2^7$ and variable batch size $b$. **f.** Forward vs the backward pass speedup of our model for fixed time steps $t = 2^7$ and variable batch size $b$. **b-f** use a 10 sample average with the mean and s.d. plotted.

## 4.1 SPEEDUP BENCHMARKS

We evaluate the speedup advantages of our model over the standard single-spike model trained using surrogate gradients, by simulating the forward and backward passes for different numbers of hidden units, layers, simulation steps and batch sizes on a synthetic spike dataset (see Appendix).

**Robust speedup for different numbers of hidden units and simulation steps**  We observe a considerable training speedup across a range of hidden units and simulation steps in a single layer (Figure 3a). We obtain an optimal speedup of 16.77× for $n = 100$ units and $t = 2^7$ time steps, where our model takes $4.34 \pm 0.9$ms compared to the $72.82 \pm 2.7$ms it takes the standard model to complete a training pass (Figure 3b). Our model still obtains a reasonable speedup of 3.40× for largest benchmarked $n = 1000$ units and $t = 2^{11}$ time steps (albeit the forward pass speedup being slower).[4]  These speedups are even more pronounced when the membrane time constants are fixed (obtaining a maximum speedup of 17.42×) or when using smaller batch sizes (with batch sizes $b = 32$ and $b = 64$ obtaining a maximum speedup of 35.05× and 25.16×, respectively; See Appendix).

**Applicability to deeper networks**  We find our model to obtain substantial training speedups when using multiple layers (Figure 3c) and layers containing thousands of neurons (Figure 3d). The training speedups remain similar across an increasing numbers of layers for different number of hidden units (Figure 3c). Furthermore, we obtain a speedup of ∼ 3× when using a large number of neurons (ranging between $2 \cdot 10^3$ to $2 \cdot 10^4$ neurons) in a single layer (Figure 3d). Inter-

---

[3]We still permit gradients to flow through the points where $g(\mathbf{z}_i^{(l)})[t] = 0$.

[4]This is due to the convolutional algorithm chosen by cudnn (Chetlur et al., 2014).

estingly, these speedups remain approximately the same across the different number of neurons, even when the batch size is changed.[5]

**Speedup advantages and room for improvement**    Previous attempts at accelerating SNN training either speed up the backward pass (Perez-Nieves & Goodman, 2021) or remove it completely (Bellec et al., 2020). These methods however still sequentially compute the forward pass, which our model is able to accelerate (Figure 3e). Furthermore, we observe the backward pass to slow down relative to the forward pass for increasing time steps (Figure 3f). Further training speedup may therefore be achieved using sparse gradient descent, as auto differentiation frameworks are not optimised for the sparse nature of SNNs (Perez-Nieves & Goodman, 2021).

## 4.2 PERFORMANCE ON REAL DATASETS

We investigate the applicability of our model to classify real data from different domains and of varying complexity (Table 1). These include the Yin-Yang dataset (Kriener et al., 2022) in which the goal is to classify spatial coordinates belonging to different groups, and the MNIST (LeCun, 1998) and Fashion-MNIST (F-MNIST) (Xiao et al., 2017) image datasets, where the objective is to classify images of handwritten digits and fashion items. All these analog datasets were converted into a spike representation using the time-to-first-spike encoding (see Appendix). We also test performance on two neuromorphic datasets, being the vision N-MNIST (Orchard et al., 2015) and the more difficult auditory SHD dataset (Cramer et al., 2020). The N-MNIST dataset is the MNIST dataset mapped onto a spike code using a neuromorphic vision sensor and the SHD dataset comprises spoken digit waveforms converted into spikes using a model of auditory bushy neurons in the cochlear nucleus.

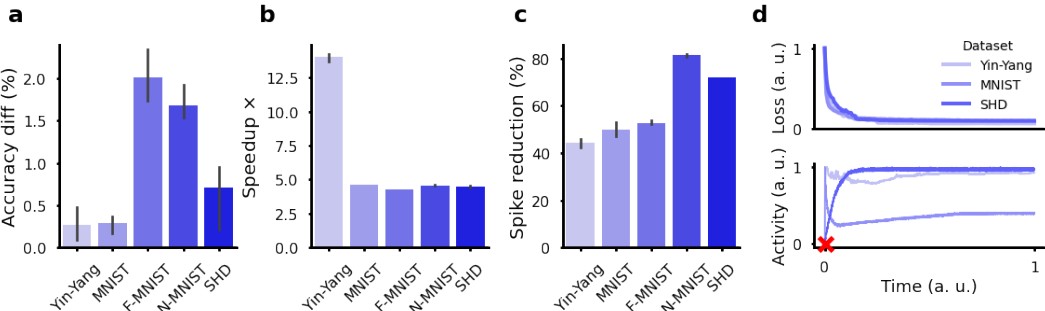

Figure 4: Analysis of our models performance on real datasets. **a.** Difference in accuracy between the standard multi-spike and our model. **b.** Training speedup of our model vs the standard single-spike model. **c.** Reduction in spikes of our single-spike model vs the standard multi-spike model (**a-c** use a 3 sample average with the mean and s.d. plotted). **d.** Training robustness of our model to solve different datasets when starting with zero network activity, which is fatal to other single-spike training methods. Top panel: Normalised training loss over time. Bottom panel: Normalised network activity over time, where the red cross denotes the absence of any spikes.

**Obtaining competitive results across different image and neuromorphic datasets**    The results of our model across all datasets are comparable or superior to prior reported results using single-spike SNNs. We reach an accuracy of 98.02%, 97.91% and 89.05% using a single hidden layer network on the Yin-Yang, MNIST and F-MNIST datasets respectively, where best performing prior work reported an accuracy of 95.90%, 98.50% and 88.1% respectively. Furthermore, our single-spike model nearly obtains the same accuracies to those obtained in the standard multi-spike SNN on these datasets (Yin-Yang and MNIST < 0.3% difference; F-MNIST ~ 2% difference; Figure 4a).

---

[5]Again, this is due to the convolutional algorithm chosen by cudnn.
[6]Results reported by Kheradpisheh et al. (2022).

Table 1: Performance comparison to existing literature (* denotes self-implementation, $^\dagger$ denotes data augmentation and $^\beta$ denotes trainable time constants).

| Dataset | Model | Spike code | Architecture | Neuron model | Accuracy (%) | Epoch time (s) |
|---|---|---|---|---|---|---|
| Yin-Yang | Göltz et al. (2021)$^\dagger$ | single | 120-10 | LIF (alpha-PSP) | $95.9 \pm 0.7$ | - |
| | **our model** | single | 120-10 | LIF$^\beta$ | $98.02 \pm 0.19$ | $1.41 \pm 0.027$ |
| | Neftci et al. (2019)* | single | 120-10 | LIF$^\beta$ | $97.91 \pm 0.37$ | $19.77 \pm 0.18$ |
| MNIST | Zhang et al. (2021)$^\dagger$ | single | 800-10 | IF (ReL-PSP) | 98.5 | - |
| | Comsa et al. (2020) | single | 340-10 | IF (alpha-PSP) | 97.90 | - |
| | **our model**$^\dagger$ | single | 1000-10 | LIF$^\beta$ | $97.91 \pm 0.10$ | $14.05 \pm 0.01$ |
| | Neftci et al. (2019)*$^\dagger$ | single | 1000-10 | LIF$^\beta$ | $97.87 \pm 0.09$ | $65.05 \pm 0.46$ |
| | Zhang et al. (2021)$^\dagger$ | single | 16C5-P2-32C5-P2-800-128-10 | IF (ReL-PSP) | 99.4 | - |
| | Zhou et al. (2021) | single | 32C5-16C5-10 | IF | 99.33 | - |
| | Mirsadeghi et al. (2021) | single | 40C5-P2-1000-10 | IF (PL-PSP) | 99.2 | - |
| | **our model**$^\dagger$ | single | 32C5-P2-64C5-P2-1000-10 | LIF$^\beta$ | $99.30 \pm 0.05$ | $23.44 \pm 0.02$ |
| | Neftci et al. (2019)*$^\dagger$ | single | 32C5-P2-64C5-P2-1000-10 | LIF$^\beta$ | $99.35 \pm 0.05$ | $32.34 \pm 0.18$ |
| FMNIST | Zhang et al. (2021)$^\dagger$ | single | 1000-10 | IF (ReL-PSP) | 88.1 | - |
| | Kheradpisheh & Masquelier (2020)$^{\dagger 6}$ | single | 1000-10 | IF | 88.0 | - |
| | **our model**$^\dagger$ | single | 1000-10 | LIF$^\beta$ | $89.05 \pm 0.27$ | $16.16 \pm 0.05$ |
| | Neftci et al. (2019)*$^\dagger$ | single | 1000-10 | LIF$^\beta$ | $89.93 \pm 0.30$ | $68.89 \pm 0.10$ |
| | Zhang et al. (2021)$^\dagger$ | single | 16C5-P2-32C5-P2-800-128-10 | IF (ReL-PSP) | 90.1 | - |
| | Mirsadeghi et al. (2021) | single | 20C5-P2-40C5-P2-1000-10 | IF (PL-PSP) | 92.8 | - |
| | **our model**$^\dagger$ | single | 32C5-P2-64C5-P2-1000-10 | LIF$^\beta$ | $90.57 \pm 0.28$ | $24.11 \pm 0.10$ |
| | Neftci et al. (2019)*$^\dagger$ | single | 32C5-P2-64C5-P2-1000-10 | LIF$^\beta$ | $90.76 \pm 0.3$ | $33.60 \pm 0.14$ |
| N-MNIST | **our model** | single | 300-10 | LIF | $95.91 \pm 0.1$ | $36.29 \pm 0.54$ |
| | | | | LIF$^\beta$ | $96.34 \pm 0.20$ | $41.93 \pm 0.42$ |
| | Neftci et al. (2019)* | single | 300-10 | LIF$^\beta$ | $97.47 \pm 0.25$ | $191.90 \pm 1.93$ |
| SHD | Cramer et al. (2020) | multi | 128-20 | LIF | $48.1 \pm 1.6$ | - |
| | Neftci et al. (2019)* | multi | 300-20 | LIF$^\beta$ | $70.81 \pm 2.05$ | $45.80 \pm 0.22$ |
| | Cramer et al. (2020) | multi | 128-20 | recurrent LIF | $71.4 \pm 1.9$ | - |
| | Perez-Nieves et al. (2021) | multi | 128-20 | recurrent LIF$^\beta$ | $82.7 \pm 0.8$ | - |
| | **our model** | single | 300-20 | LIF | $44.50 \pm 2.65$ | $8.68 \pm 0.01$ |
| | | | | LIF$^\beta$ | $70.32 \pm 0.30$ | $11.27 \pm 0.21$ |
| | Neftci et al. (2019)* | single | 300-20 | LIF$^\beta$ | $68.91 \pm 0.25$ | $50.12 \pm 0.13$ |

**Single-spike neurons solve challenging temporal problems using neural heterogeneity** It has been noted that single-spike SNNs are well suited for static datasets (such as spike encoded images) and less suited for processing temporally complex stimuli (such as audio or video) due to the single-spike constraint (Zenke et al., 2021; Eshraghian et al., 2021). Prior single-spike SNN training techniques have attempted to optimise network connectivity without learning other neural parameters, such as membrane time-constants, which have shown to improve performance in multi-spike SNNs (Perez-Nieves et al., 2021). We explored the effect of learning the membrane time-constants in our single-spike model. We obtained an accuracy of 44.50% using a network trained with fixed time constants on the temporally-complex auditory SHD dataset. However, by including learnable time constants we were able to obtain a much higher accuracy of 70.32%, which is similar to the performance obtained by a standard SNN with trainable time constants 70.81% or recurrent connections 71.40%.

**Drastic speedup in training** We obtain over a four-fold training speedup across all datasets over the standard single-spike SNN, with a maximum speedup of 13.98× on the Yin-Yang dataset (Figure 4b). We observe similar training speedups over the multi-spike SNN (see Appendix). Differences in speedups are due to the different temporal lengths and input dimensions of the datasets, as well the different network architectures employed (see section 4.1).[7]

**Increased spike sparsity** Our single-spike SNN is able to solve various datasets with a large reduction in spikes compared to a standard multi-spike SNN (Figure 4c), with over a 44% and up to a 81% reduction in spikes. This corroborates the value of obtaining more energy-efficient computations using single- rather than multi-spike neuromorphic systems (Liang et al., 2021; Oh

---

[7]Note that - unlike the neuromorphic datasets - the training speedup for the image datasets is dependent on the selected number of simulation time steps for transforming an image into the temporal domain (see Appendix for chosen values).

et al., 2021; Zhou et al., 2021), as energy consumption scales approximately proportional to the number of emitted spikes (Panda et al., 2020).

**Training deeper convolutional architectures**   We evaluate our model in deeper convolutional architectures, which to date remains largely unexplored in single-spike SNNs (Mirsadeghi et al., 2022). We trained a multi-layer convolutional network on the MNIST and F-MNIST datasets, obtaining accuracies (MNIST: 99.32% and F-MNIST: 90.57%) similar to best performing prior work (MNIST: 99.4% and F-MNIST: 92.8%), whilst being faster to train in comparison to the control (MNIST-speedup ~ 1.37× and F-MNIST-speedup ~ 1.39×).

**Robust learning and bypassing the dead neuron problem**   A limitation of current single-spike SNN training methods is the dead neuron problem, referring to the hinderance in learning when neurons do not spike, as the learning signal is dependent on the occurrence thereof (Eshraghian et al., 2021). Our model is able to overcome this problem as we use surrogate gradients for training, in which the learning signal is instead passed through the membrane potentials. We experimentally verified this by showing how networks instantiated with zero starting activity (fatal to other single-spike training methods) still manage to solve different datasets (Figure 4d).

## 5   DISCUSSION

SNNs emulated on neuromorphic hardware are a promising avenue towards addressing the energy and scaling constraints of ANNs (Wunderlich et al., 2019). Single-spike SNNs further amplify these energy improvements through extreme spike sparsity, as energy consumption scales approximately proportionally to the number of emitted spikes (Panda et al., 2020). To date, SNN training remains challenging due to the non-differentiable nature of the spike function, prohibiting the direct use of the backprop training algorithm which underpins the success of ANNs. Various extensions of backprop for SNNs have been proposed, but fall short in particular aspects. Gradients can be passed through the timing of spikes (Bohte et al., 2002; Mostafa, 2017; Kheradpisheh & Masquelier, 2020), yet this method suffers from the dead neuron problem, requires careful regularisation or imposes computationally-expensive modelling constraints. Alternatively, gradients can be passed through the membrane potentials using surrogate gradients (Shrestha & Orchard, 2018; Neftci et al., 2019), and although this method improves upon the problems of passing gradients through the spike times, it is painfully slow.

In this work, we address these problems by proposing a new general model (e.g. neurons can be IF or LIF) for training single-spike SNNs, without imposing any modelling (e.g. requiring PSP kernels) or training constraints (e.g. requiring careful regularisation) and support training of neural parameters other than synaptic connectivity (e.g. membrane time constants). We mathematically show how training can be sped up by replacing the slow sequential operations with faster convolutional ones. We experimentally validate this speedup across various numbers of units, time steps, layers and batch sizes, obtaining up to a 16.77× speedup. We show that our model can be trained across different network architectures (e.g. feedforward, hierarchical and convolutional) and obtain competitive results on different image and neuromorphic datasets. Our results compare well against multi-spike SNNs (< 2% accuracy difference on all datasets) and obtain up to an 81% reduction in spike counts. Furthermore, our method circumvents the dead neuron problem and, for the first time, we show how single-spike SNNs can solve temporally-complex datasets on a par with multi-spike SNNs by including trainable membrane time constants. Our findings therefore challenge the dogma that single-spike SNNs are only suited to non-temporal problems (Eshraghian et al., 2021; Zenke et al., 2021).

We obtain training speedups on all datasets, however, find that the backward pass slows down relative to the forward pass for longer timespans. Future work could mitigate this bottleneck and accelerate training using sparse gradient descent, which has shown to accelerate the backward pass in standard SNNs by taking advantage of spike sparsity (Perez-Nieves & Goodman, 2021). Currently, our single-spike model performs slightly worse compared to its multi-spike counterpart, where better performance could be achieved by extending our model to the multi-spike setting and permitting recurrent connectivity. Finally, it remains an open question how the inclusion of trainable membrane time constants in our model boost performance, requiring further theoretical analysis.

## 6 Reproducibility statement

The theoretical construction and derivations of our model are outlined in section 3 and we provide accompanying derivations in the Appendix. All code is publicly available at https://github.com/webstorms/Block under the BSD 3-Clause Licence. This includes instructions on installation, data processing and running experiments to reproduce all results and figures portrayed in the paper. Training details are also provided in the Appendix.

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

# A  APPENDIX

## A.1  SPIKING NEURAL NETWORK DERIVATIONS

### A.1.1  NORMALISING THE LEAKY INTEGRATE AND FIRE MODEL

**Proposition 2.** *Any leaky integrate and fire (LIF) model* $\tau \frac{dV(t)}{dt} = -V(t) + V_{rest} + RI(t)$ *(with membrane potential $V$, resting potential $V_{rest}$, firing threshold $V_{th}$, resistance $R$, input current $I$ and membrane time constant $\tau$) can be normalised to a LIF model of the form* $\tau \frac{d\tilde{V}(t)}{dt} = -\tilde{V}(t) + \tilde{I}(t)$ *(such that $0 \leq \tilde{V}(t) \leq 1$, with firing threshold $\tilde{V}_{th} = 1$, resting potential $\tilde{V}_{rest} = 0$ and a resistance equal to one).*

*Proof.* This mapping from any LIF model to the normalised LIF model is achieved using the following transformation (taken from Hunsberger (2018)).

$$\tilde{V}(t) = \frac{V(t) - V_{rest}}{V_{th} - V_{rest}} \tag{9}$$

Rearranging this expression with respect to $V(t) = \tilde{V}(t)(V_{th} - V_{rest}) + V_{rest}$ and substituting this into the LIF model we obtain

$$
\begin{aligned}
\tau \frac{dV(t)}{dt} &= -V(t) + V_{rest} + RI(t) \\
(V_{th} - V_{rest})\tau \frac{d\tilde{V}(t)}{dt} &= -\Big( \tilde{V}(t)(V_{th} - V_{rest}) + V_{rest} \Big) + V_{rest} + RI(t) \\
\tau \frac{d\tilde{V}(t)}{dt} &= -\tilde{V}(t) + \underbrace{\frac{R}{V_{th} - V_{rest}} I(t)}_{\text{Input current } \tilde{I}(t)}
\end{aligned}
\tag{10}
$$

This new LIF form has a resting potential $\tilde{V}_{rest} = 0$ and firing threshold $\tilde{V}_{th} = 1$ (obtained by substituting $V(t) = V_{rest}$ and $V(t) = V_{th}$ in Equation 9 respectively). Thus, without loss of generality, any LIF model can be mapped to a normalised form using linear transformation Equation 9. □

### A.1.2  DISCRETISING THE LEAKY INTEGRATE AND FIRE MODEL

**Proposition 3.** *The normalised continuous time leaky integrate and fire model* $\tau \frac{dV(t)}{dt} = -V(t) + I(t)$ *(with membrane potential $V$, input current $I$ and membrane time constant $\tau$) can numerically be approximated by discrete time difference equation $V[t+1] = \beta V[t] + (1-\beta)I[t+1]$, where $\beta = \exp(\frac{\Delta t}{\tau})$ (for simulation time resolution $\Delta t$).*

*Proof.* We proceed using the forward Euler method. Let $I(t) = I$ be constant with respect to time, for which the ordinary differential equation becomes separable.

$$
\begin{aligned}
\tau \frac{dt V(t)}{dt} &= -V(t) + I \\
\int \frac{dV}{V(t) - I} &= -\frac{1}{\tau} \int dt \\
\ln(V(t) - I) &= -\frac{1}{\tau} t + \ln(k) \\
V(t) &= k \exp(-\frac{1}{\tau} t) + I
\end{aligned}
\tag{11}
$$

For initial solution $V(t_0)$ at time $t_0$ we derive $k = (V(t_0) - I)\exp(\frac{t_0}{\tau})$. Then for constant $I$ and initial solution $V(t_0)$ we obtain solution.

$$
\begin{aligned}
V(t) &= (V(t_0) - I)\exp(-\frac{t - t_0}{\tau}) + I \\
&= \exp(-\frac{t - t_0}{\tau})V(t_0) + (1 - \exp(-\frac{t - t_0}{\tau}))I
\end{aligned}
\tag{12}
$$

To obtain the discretised update equation, we define simulation update time step $\Delta t = t - t_0$, decay factor $\beta = \exp(-\frac{\Delta t}{\tau})$, assign continuous time points to discretised time steps $t \leftarrow t_0$ and $t + 1 \leftarrow t_0 + \Delta t$ and assume the input current to be approximately constant and equal to $I[t + 1]$ between discretised update steps $t$ to $t + 1$.

$$
V[t + 1] = \beta V[t] + (1 - \beta)I[t + 1]
\tag{13}
$$

$\square$

### A.1.3 UNROLLING THE LEAKY INTEGRATE AND FIRE MODEL WITHOUT THE RESET TERM

**Proposition 4.** *Equation* $V[t] = \beta^t V[0] + (1 - \beta)\sum_{i=1}^{t}\beta^{t-i}I[i]$ *is equivalent to difference equation* $V[t] = \beta V[t - 1] + (1 - \beta)I[t]$ *for* $t \geq 1$.

*Proof.* We proceed to proof equivalence by induction. For $t = 1$ we obtain

$$
\begin{aligned}
V[1] &= \beta^1 V[0] + (1 - \beta)\sum_{i=1}^{1}\beta^{1-i}I[i] \\
&= \beta^1 V[0] + (1 - \beta)I[1]
\end{aligned}
\tag{14}
$$

Hence the relation holds true for the base case $t = 1$. Assume the relation holds true for $t = k \geq 1$, then for $t = k + 1$ we derive

$$
\begin{aligned}
V[k + 1] &= \beta V[k] + (1 - \beta)I[k + 1] \\
&= \beta\Big(\beta^k V[0] + (1 - \beta)\sum_{i=1}^{k}\beta^{k-i}I[i]\Big) + (1 - \beta)I[k + 1] \\
&= \beta^{k+1} V[0] + (1 - \beta)\sum_{i=1}^{k}\beta^{(k+1)-i}I[i] + (1 - \beta)I[k + 1] \\
&= \beta^{k+1} V[0] + (1 - \beta)\sum_{i=1}^{k+1}\beta^{(k+1)-i}I[i]
\end{aligned}
\tag{15}
$$

This implies equivalence for $t = k+1$ if $t = k$ holds true. By the principle of induction, equivalence is established given that both the base case and inductive step hold true.

$\square$

### A.2 ADDITIONAL MODEL THEORY: WHY IS OUR MODEL FASTER?

To address the question why our model is faster than the standard single-spike model, we analyse their respective computational complexities. Consider a single neuron with $N$ presynaptic neurons simulated for $T$ time steps. Our model has a computational complexity of $O(NT^2)$ and the computational complexity of the standard model is $O(NT)$. However, the sequential complexity of our model is constant time $O(1)$ (as our model eschews all sequential dependence) and the sequential complexity of the standard model is linear $O(T)$. Our model performs more calculations than the standard single-spike model, yet is able to obtain faster training speeds, as - unlike the standard single-spike model - all these calculations are highly parallelisable.

### A.3  ADDITIONAL DATASET DETAILS

#### A.3.1  SYNTHETIC SPIKE DATASET FOR THE SPEED BENCHMARKS

We generated binary input spike tensors of shape $B \times N \times T$ ($B$ being the batch size, $N$ the number of input neurons and $T$ the number of simulation steps). For every batch dimension $b$ a firing rate $r_b \sim \mathbf{U}(u_{\min}, u_{\max})$ was uniformly sampled (with $u_{\min} = 0$Hz and $u_{\max} = 200$Hz), from which a random binary spike matrix of shape $N \times T$ was constructed, such that every input neuron in this matrix had an expected firing rate of $r_b$Hz.

#### A.3.2  TIME-TO-FIRST-SPIKE ENCODING

We encoded all analog non-spiking input data into a spike raster using the time-to-first-spike coding method (Kheradpisheh & Masquelier, 2020). Here, every scalar value $I_i \in \{0, I_{\max}\}$ within an input tensor is converted into a spike train with a single spike, where the time of spike $t_i \in \{0, T\}$ is determined by the following equation

$$t_i = \lfloor \frac{I_{\max} - I_i}{I_{\max}} T \rfloor \tag{16}$$

### A.4  TRAINING DETAILS AND HYPERPARAMETERS

#### A.4.1  READOUT NEURONS

The output layer $L$ of every trained network contained the same number of neurons as the number of classes contained within the dataset being trained on. As suggested by Zenke & Vogels (2021), every neuron had a firing threshold set to infinity (*i.e.* the spiking and reset mechanism was removed) from which the output $o_{b,c}$ of readout neuron $c$ to input sample $b$ was either taken to be the maximum membrane potential over time $o_{b,c} = \max_t V_{b,c}^L[t]$ or the summated membrane potential over time $o_{b,c} = \sum_t V_{b,c}^L[t]$ (see table 2).

#### A.4.2  BETA CLIPPING

As the beta $\beta_i^{(l)}$ (a transformation of the membrane time constant) of every neuron was optimised, we had to enforce correct neuron dynamics by clipping the values into the range $[0, 1]$. Note that $\beta_i^{(l)} = 0$ implies no memory i.e. a binary neuron, $0 < \beta_i^{(l)} < 1$ implies decaying memory i.e. a LIF neuron and $\beta_i^{(l)} = 1$ implies full memory i.e. an IF neuron.

$$\beta_i^{(l)} = \begin{cases} 1, & \text{if } \beta_i^{(l)} > 1 \\ 0, & \text{if } \beta_i^{(l)} < 0 \end{cases} \tag{17}$$

#### A.4.3  WEIGHT INITIALISATION

The network weights in a layer were sampled from a uniform distribution $\mathbf{U}(-\sqrt{N^{-1}}, \sqrt{N^{-1}})$, except for the Yin-Yang dataset for which the weights were sampled from $\mathbf{U}(-\sqrt{2N^{-1}}, \sqrt{2N^{-1}})$. For the feedforward layers $N$ was set to the number of afferent connections to the layer and for the convolutional layers $N = k^2$ for kernel shape $k \times k$. The bias terms were initialised to 0 in all networks. All neurons in the hidden layers were initialised with a membrane time constant $\tau = 10$ms and $\tau = 20$ms for the readout neurons.

#### A.4.4  SUPERVISED TRAINING LOSS

All networks were trained to minimise a cross-entropy loss

$$\mathcal{L} = -\frac{1}{B} \sum_{b=1}^{B} \sum_{c=1}^{C} y_{b,c} \log(p_{b,c}) \tag{18}$$

with $B$ and $C$ being the number of batch samples and dataset classes respectively, and $y_{b,c} \in \{0,1\}^C$ and $p_{b,c}$ being the one hot target vector and network prediction probabilities respectively. The prediction probabilities $p_{b,c}$ were obtained by passing the readout neuron outputs $o_{b,c}$ through the softmax function.

$$p_{b,c} = \frac{\exp o_{b,c}}{\sum_{k=1}^{C} \exp o_{b,k}} \tag{19}$$

### A.4.5 SURROGATE GRADIENT

The backprop algorithm requires all nodes within the computational graph of optimisation to be differentiable. This requirements is however violated in a SNN due to the non-differentiable heavy stepwise spike function $f$. To permit the use of backprop, we replaced the undefined derivate $\frac{df}{dV}$ of the function $f$ with a surrogate gradient $\frac{df_{\text{sur}}}{dV}$ (Zenke & Ganguli, 2018), which has been shown to work well in practice (Zenke & Vogels, 2021). Here hyperparameter $\beta_{\text{sur}}$ (which we set to 10 in all experiments) defines the slope of the gradient.

$$\frac{df_{sur}(V)}{dV} = (\beta_{\text{sur}}|V| + 1)^{-2} \tag{20}$$

### A.4.6 TRAINING PROCEDURE

All models were trained using the Adam optimiser (with default parameters) (Kingma & Ba, 2014). Training started with an initial learning rate, which was decayed by a factor of 10 every time the number of epochs reached a new milestone, after which the best performing model (that achieved lowest training loss) was loaded and training continued.

### A.4.7 TRAINING HYPERPARAMETERS

Table 2: Dataset and corresponding training parameters.

|  | Yin-Yang | MNIST | conv MNIST | F-MNIST | conv F-MNIST | N-MNIST | SHD |
|---|---|---|---|---|---|---|---|
| Dataset (train/test) | 20k/10k | 60k/10k | 60k/10k | 60k/10k | 60k/10k | 60k/10k | 8156/2264 |
| Input neurons | 4 | 784 | $28 \times 28$ | 784 | $28 \times 28$ | 1156 | 700 |
| Dataset classes | 3 | 10 | 10 | 10 | 10 | 10 | 20 |
| Epochs | 200 | 140 | 140 | 140 | 140 | 200 | 200 |
| Learning rate | 0.001 | 0.001 | 0.001 | 0.001 | 0.001 | 0.0002 | 0.0002 |
| Batch size $B$ | 128 | 128 | 64 | 128 | 64 | 128 | 128 |
| Simulation steps $T$ | 100 | 100 | 8 | 100 | 8 | 300 | 500 |
| Time resolution $\Delta t$ (ms) | 1 | 1 | 1 | 1 | 1 | 1 | 2 |
| Milestones | $(50, 100)$ | $(15, 90, 120)$ | $(30, 60, 90)$ | $(15, 90, 120)$ | $(30, 60, 90)$ | $(30, 60, 90)$ | $(30, 60, 90)$ |
| Output function | sum | sum | max | sum | sum | sum | sum |

### A.5 EXTENDED SPEEDUP RESULTS

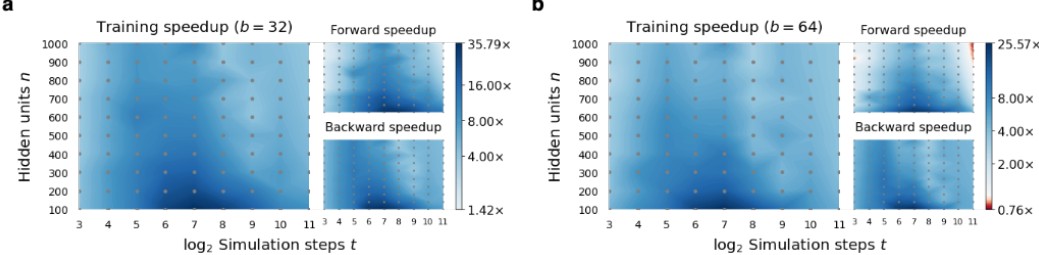

Figure 5: Total training speedup using smaller batch sizes as a function of the number of hidden neurons $n$ and simulation steps $t$ (left), alongside the corresponding forward and backward pass speedups (right). **a.** Speedups using batch size $b = 32$. **b.** Speedups using batch size $b = 64$.

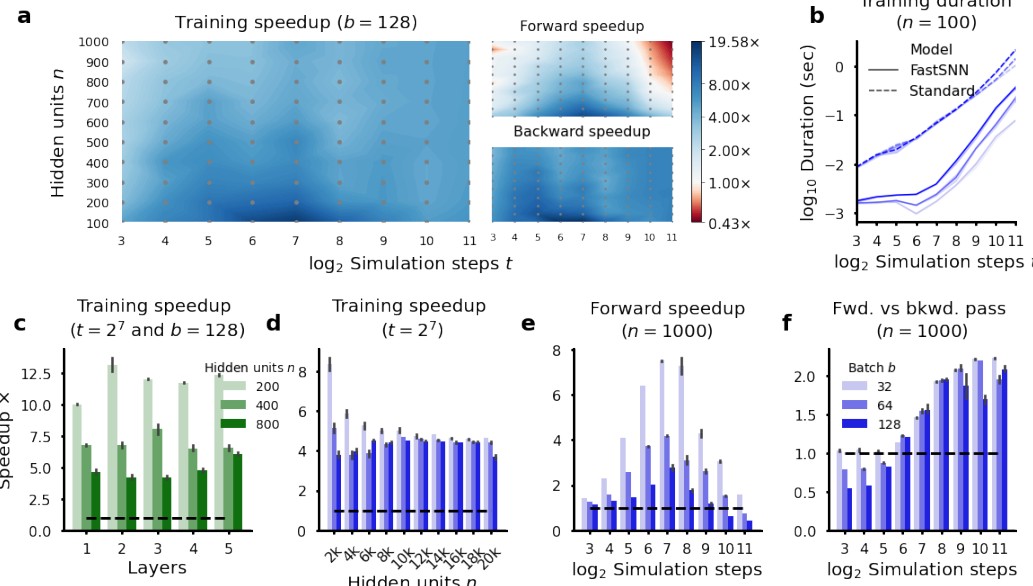

Figure 6: Training speedup of our model over the standard model (using fixed membrane time constants). **a.** Total training speedup as a function of the number of hidden neurons $n$ and simulation steps $t$ (left), alongside the corresponding forward and backward pass speedups (right). **b.** Training durations of both models for fixed hidden neurons $n = 100$ and variable batch size $b$. **c.** Training speedup over different number of layers for fixed time steps $t = 2^7$ and batch size $b = 128$. **d.** Training speedup over large number of hidden neurons $n$ for fixed time steps $t = 2^7$ and variable batch size $b$. **e.** Forward pass speedup for fixed time steps $t = 2^7$ and variable batch size $b$. **f.** Forward vs the backward pass speedup of our model for fixed time steps $t = 2^7$ and variable batch size $b$. **b-f** use a 10 sample average with the mean and s.d. plotted.

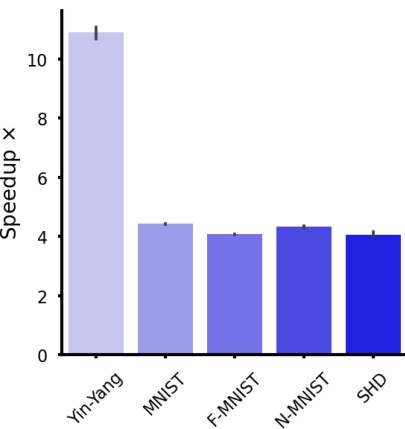

Figure 7: Training speedup of our model vs the standard multi-spike model across different datasets (using a 3 sample average with the mean and s.d. plotted)

