# OpenReview forum: "Robust and accelerated single-spike spiking neural network training with applicability to challenging temporal tasks"
_ICLR.cc/2023/Conference — Submitted to ICLR 2023_

### Official Review · Reviewer_5huF · 2022-10-24

**Confidence:** 3
**Clarity, Quality, Novelty And Reproducibility:** mentioned above.
**Correctness:** 3
**Technical Novelty And Significance:** 3
**Empirical Novelty And Significance:** 3
**Recommendation:** 5

**Strength And Weaknesses:**

mentioned above.

**Summary Of The Paper:**

This paper presented an interesting idea for accelerating the training procedure of SNNs. The proposed SNN is considered to reduce consumption by enabling SNNs to maintain at most one firing spike.

The organization of this paper is clear, and citations are also appropriate. The experiments conducted on various datasets verify the effectiveness of the proposed method.

Major Issuses:
1. As claimed in this paper, the single-spike model can speed up the training procedure of SNNs due to a decreasing number of firing spikes. Thus, it is natural to verify the computational complexity of the single-spike and multi-spike models, instead of running time. In other words, the running time is not sufficient, and the computational complexity is very likely to be the same order of magnitude.
2. As shown in this paper, when the authors reduce the number of firing spikes, the single-spike neural network not only has a faster training procedure, but also performs other comparative models, even avoiding the problem of "dead neurons". According to Occam's Razor, what is the price of superior performance? It is significant to be discussed in this work.

**Summary Of The Review:**

Overall, I believe this is an interesting paper that focuses on an energy saving way of SNNs. However, I still find that there is something has been hidden by the authors, which urgently need to be analyzed and discussed in depth. Therefore, I tend to accept this paper if the authors fixed these issues in the next phase.

---

> ### Author Response · Authors · 2022-11-18
> **Response to Reviewer 5huF**
>
> Thank you for taking the time to review our submission. We hope our responses address your concerns and we welcome any further questions you may have.
>
> **As claimed in this paper, the single-spike model can speed up the training procedure of SNNs due to a decreasing number of firing spikes.**
>
> To expand on this (and to clarify), the training speedup is not dependent on the number of spikes elicited. Rather, we constrain neurons to spike at most once, which thus allows us to evolve neuron dynamics using faster convolutional operations, rather than slower sequential ones.
>
> **Thus, it is natural to verify the computational complexity of the single-spike and multi-spike models, instead of running time. In other words, the running time is not sufficient, and the computational complexity is very likely to be the same order of magnitude.**
>
> You raise an interesting point and we thank you for your insightful question! The computational complexity of our model is actually worse than the standard model. Consider a single neuron with N presynaptic neurons simulated for T time steps. Our model has a computational complexity of $O(NT^2)$ vs the computational complexity $O(NT)$ of the standard model (the extra computational factor T in our model is due to the convolution with a kernel of length T, denoted as bold $\mathcal{\beta}$ in our paper in equation 5). However, due to the mathematical construction of our model, the sequential complexity of our model is constant $O(1)$ vs the linear dependence on time $O(T)$ of the standard model. Thus, all calculations in our model are readily parallelisable (hence the drastic training speedup when training on modern GPUs). We have updated the paper to emphasise these points (brief mention in section 3 and more thorough explanation in Appendix A.2).
>
> **As shown in this paper, when the authors reduce the number of firing spikes, the single-spike neural network not only has a faster training procedure, but also performs other comparative models, even avoiding the problem of "dead neurons". According to Occam's Razor, what is the price of superior performance? It is significant to be discussed in this work.**
>
> The price of superior performance (with respect to training speed) is carrying out more - yet highly parallelisable - computational work as just discussed. The price for the high accuracy on the challenging neuromorphic SHD dataset is to include the membrane time constants into the computational graph, which slightly slows down the training speed in our model: With fixed membrane time constants our model trains in 8.68 seconds per epoch and reaches an accuracy of 44.5%, whilst with trainable membrane time constants our model trains in 11.27 seconds per epoch, yet reaches an accuracy of 70.32%.
>
> **Overall, I believe this is an interesting paper that focuses on an energy saving way of SNNs. However, I still find that there is something has been hidden by the authors, which urgently need to be analyzed and discussed in depth. Therefore, I tend to accept this paper if the authors fixed these issues in the next phase.**
>
> We thank you for being open minded and hope our answers clarify your concerns. We are happy to answer any more questions you may have or go into more depth in our answers if you like. Rather than hiding any details, we have published all code [online](https://github.com/webstorms/Block) for others to replicate our findings. In addition, we have written a short [tutorial]( https://github.com/webstorms/Block/blob/main/notebooks/Tutorial.ipynb) to help others get started with our model.

---

> > ### Comment · Reviewer_5huF · 2022-11-20
> > **Responses**
> >
> > After reading the other reviewers comments and authors' response, I have a negative comment on this paper. Of course, the speedup does not depend on the number of exciting spikes. In contrast to the conventional methods that speed up the training procedure of SNNs by approximating the non-linear and non-differential firing functions, suppressing the neural excitation in this work would avoid recursive computing and decrease the number of non-linear activation computations, which allow some speedup operations, such as parallel computing, convolution operation, etc. However, if so, the performance of single-spike SNNs cannot exceed, or even transcend that of ANNs or DNNs, since almost no neural excitation is equivalent to ensuring neural excitation of almost all neurons at any timestamps. In addition, this approach has many disadvantages:
> > (1) The spiking neuron model is no longer capable of spatiotemporal representation at the unit level. When the spike is not excited, the neural computing is just like a special formation of the conventional MP model.
> > (2) This practice is not biologically plausible.
> > (3) It is still significant to explore the killer applications, e.g., some neuromorphic datasets with long temporal lengths, of SNNs, although one usually employs the image data sets with neural encoding as the benchmarks. Thus, discussing the length of simulation sampling on MNIST does not excite me.
> >
> > Regarding the significance of this paper, this paper does not care about spike calculations in essence, but simplifies spike calculations into a special MP case, and thus I think its significance is also limited. It is unlikely to make a huge impact. Therefore, I do not recommend accepting this paper.

---

### Official Review · Reviewer_cHb7 · 2022-10-25

**Confidence:** 4
**Correctness:** 3
**Technical Novelty And Significance:** 2
**Empirical Novelty And Significance:** 2
**Recommendation:** 3

**Clarity, Quality, Novelty And Reproducibility:**

The paper is well written and easy to follow. It provides a novel approach to decrease the spiking rate.

**Strength And Weaknesses:**

Strength：
1. This material is generally well-presented and easy to read.
2. It’s a good idea to speed up the SNN training using parallel acceleration.

Weakness:
1. When comparing accuracy, acceleration, and spike reduction, specific corresponding baseline models must be provided.
2. The simulation time steps in the experiments are usually higher than the required value. E.g., MNIST datasets may only need 8 simulation time steps, the same as conv MNIST. When the simulation time steps are small, the training acceleration of this method in the paper is not so significant as the authors claim.
3. The article's novelty and contribution are modest, and it offers few enthralling or motivating insights.

Questions:
1. Which part of the paper's algorithm is robust enough to declare in the title?
2. In Table 1, most of the experimental results are worse than the existing methods, especially on more complex convolutional networks. So, I doubt whether the authors' method can be implemented into mainstream deeper convolutional to take full advantage of the training efficiency.


**Summary Of The Paper:**

This paper unrolls the update formula of the LIF model along the time axis and uses the vectorized variables for parallel acceleration. It proposes a simple method to determine when the first spikes are fired. In this manner, both the goal of speeding up training and lowering the spike counts are accomplished.

**Summary Of The Review:**

Overall, this paper proposes an interesting idea to speed up the training SNN  through parallel acceleration. Based on the description, I think the acceleration is real but the deficit of representation power of large models remain concerned as well as its advantage of the quantized model.

---

> ### Author Response · Authors · 2022-11-18
> **Response to Reviewer cHb7 (Part 1)**
>
> Thank you for taking the time to review our submission. We hope our responses address your concerns and we welcome any further questions you may have.
>
> **When comparing accuracy, acceleration, and spike reduction, specific corresponding baseline models must be provided.**
>
> We are happy to include more baseline models upon your request and address any questions regarding the control experiments performed. We compared the training speedup of our model to a standard single-spike SNN trained using surrogate gradients and calculated the spike reduction of our model to a standard multi-spike SNN also trained using surrogate gradients ([Neftci et al. (2019)]( <https://ieeexplore.ieee.org/stamp/stamp.jsp?arnumber=8891809>)). Across the different datasets we compared the accuracy of our single-spike model to other single-spike SNNs found in the literature (Table 1).
>
> **The simulation time steps in the experiments are usually higher than the required value. E.g., MNIST datasets may only need 8 simulation time steps, the same as conv MNIST. When the simulation time steps are small, the training acceleration of this method in the paper is not so significant as the authors claim.**
>
> The training acceleration over longer simulation time steps is of particular interest for neuromorphic datasets (which have long temporal lengths) and -as you have pointed out- of less interest to image datasets (as here the simulation duration is usually arbitrarily chosen, and good performance can be achieved with a few time steps). We have updated the paper with a footnote on page 8 to clarify and emphasise this.
>
> **Which part of the paper's algorithm is robust enough to declare in the title?**
>
> 1. Robust training. Our model avoids training instabilities such as the dead neuron problem (where lack of spike activity can otherwise halt learning in other single-spike SNN training approaches).
> 2. Architectural robustness. Our method is applicable to different types of network architectures (e.g. feedforward, convolutional and multiple layers).
> 3. Robust performance. For the first time we show single-spike SNNs to perform well on challenging neuromorphic datasets such as the SHD dataset (performing on a par with multi-spike SNNs).
>
> **In Table 1, most of the experimental results are worse than the existing methods, especially on more complex convolutional networks.**
>
> Our primary focus was to accelerate the training of single-spike SNNs using surrogate gradients and to tackle challenging temporal neuromorphic datasets, which we successfully managed to do (at least a x4 training speedup on all datasets over the control and SOTA accuracy on the neuromorphic Yin-Yang and SHD datasets). We obtain comparable performance - albeit slightly worse - on the image datasets (MNIST, our model: 99.3% vs best other model: 99.4%; F-MNIST, our model: 90.57% vs best other model: 92.8%). We have included these results to demonstrate the versatility of our method to different problem domains (i.e. image and neuromorphic datasets, where prior single-spike SNNs just focus on image datasets) and to different network architectures (e.g. feedforward, convolutional and multiple layers). We never set out to obtain SOTA accuracies on image classification, but to rather accelerate single-spike SNN training over long timespans, and to apply these networks to - as of yet unexplored - neuromorphic datasets.

---

> > ### Author Response · Authors · 2022-11-18
> > **Response to Reviewer cHb7 (Part 2)**
> >
> > **So, I doubt whether the authors' method can be implemented into mainstream deeper convolutional to take full advantage of the training efficiency.**
> >
> > These are important concerns. However, as you have pointed out, image classification using convolutional SNNs do not require many simulation steps, and as our training speedup is over the temporal domain, our method would not take full advantage of its training efficiency when using deeper convolutional architectures (at least for image classification). Yet perhaps there is a problem which requires a deep convolutional SNN trained over long timespans. At least to us it remains unclear what problems these could be  (in the context of problems currently explored using SNNs). Image datasets are readily classified using large SNN networks with few time steps and neuromorphic datasets are readily classified using small networks with many time steps.
> >
> > However you raise an important - and as of yet - unsolved problem: Training deep SNNs over large number of time steps. At least using surrogate gradient training (the status quo for directly training SNNs) there are inherent memory constraints which would prohibit this. In its conventional use, surrogate gradient training is a form of backprop through time (BPTT), for which a copy of the entire network state needs to be retained in memory at every simulation time step ([Neftci et al. (2019)](<https://ieeexplore.ieee.org/stamp/stamp.jsp?arnumber=8891809>)). Although recent work has managed to reduce these memory requirements ([Perez-Nieves & Goodman (2021)](https://proceedings.neurips.cc/paper/2021/file/61f2585b0ebcf1f532c4d1ec9a7d51aa-Paper.pdf)), memory still poses a non-trivial problem for scaling SNNs to deep networks trained over many time steps. In our model we mitigate the dependence on BPPT, as we replace all sequential operations with faster convolutional ones. However, the temporal length of the kernels which we employ are of equal length to the number of simulation steps. Thus, a trade-off between network size and number of simulation steps must still be made to fit all model weights into GPU memory.
> >
> > Following your concerns - and to explore the applicability of our model to deeper networks - we have trained a 7 layer spiking convolutional VGG network using our method on the CIFAR10 dataset, and obtained an accuracy of 82.14% (see [notebook](https://github.com/webstorms/Block/blob/main/notebooks/review/VGG%20CIFAR10%20.ipynb)). In comparison, better results have been reported in a 16 layer convolutional VGG network (92.36%) ([Zhou et al. (2021)](https://ojs.aaai.org/index.php/AAAI/article/view/17329)). To emphasise, the aim of our work was not to obtain SOTA on image classification, but rather to accelerate single-spike SNN training over long time spans and to obtain improved performance on - more applicable - neuromorphic datasets (which we successfully managed). At the moment, there is no single-spike training method that is applicable to both large scale image datasets and to temporally long and challenging neuromorphic datasets. The single-spike method developed by ([Zhou et al. (2021)](https://ojs.aaai.org/index.php/AAAI/article/view/17329)) is more applicable for training large convolutional SNNs for image classification and our method is more applicable for training smaller SNNs for temporal neuromorphic datasets.
> >
> > **Overall, this paper proposes an interesting idea to speed up the training SNN through parallel acceleration. Based on the description, I think the acceleration is real but the deficit of representation power of large models remain concerned as well as its advantage of the quantized model.**
> >
> > Thank you - We hope we were able to address your concerns and clarify why the training of large SNNs over many time steps remains an open problem. Our work has focused on improving training of smaller SNNs over many time steps (where we successfully demonstrate our new single-spike model to obtain fast training speeds and SOTA on challenging neuromorphic datasets). Other work has focused on training large scale SNNs for image classification, yet on a few time steps (and are less suited for challenging neuromorphic datasets). No method - as of yet - manages to do both. We believe our work to be of interest to the SNN community and hope that future investigations manage to scale our model.

---

> > > ### Comment · Reviewer_cHb7 · 2022-11-20
> > > **Response to the rebuttal**
> > >
> > > Thank you for answering my questions. I have read the response to my concerns as well as to the other reviewers. My major concern is not whether the proposed method is effective in its provided experiments but how extendible it is on complex datasets or scenarios, which is not included in the current manuscript. It is not convincing enough that the current setup is promising for the SNN field and it is necessary to identify the killer application to support the efficacy of the proposed method. Thus, I decided to keep my scores.

---

### Official Review · Reviewer_HU9D · 2022-10-25

**Confidence:** 4
**Clarity, Quality, Novelty And Reproducibility:** Nothing to declare.
**Correctness:** 3
**Technical Novelty And Significance:** 2
**Empirical Novelty And Significance:** 2
**Recommendation:** 3

**Strength And Weaknesses:**

STRENGTHS:

TTFS is appealing because it consumes few spikes, therefore little energy

WEAKNESSES:

The authors limit themselves to extremely simple toy datasets (not even CIFAR!), and even on these datasets, the method is not competitive w.r.t. other proposals (see Table 1).
Zhou et al AAAI 2021 should be included in Table 1. They have decent results on CIFAR, and even on ImageNet, using TTFS!



**Summary Of The Paper:**

The authors use spiking neural networks for image classification. They use time-to-first-spike (TTFS) coding. After a neuron has fired, any eventual subsequent spike is blocked. Training is done with a surrogate gradient.

**Summary Of The Review:**

There is a new method for training TTFS-SNNs, but it's unclear what the advantages are w.r.t. existing methods.

---

> ### Author Response · Authors · 2022-11-18
> **Response to Reviewer HU9D (Part 1)**
>
> Thank you for taking the time to review our submission. We hope our responses address your concerns and we welcome any further questions you may have.
>
> **The authors use spiking neural networks for image classification.**
>
> The image classification is a smaller part of our investigation, which we included to demonstrate the versatility of our model to different problem domains and model architectures. Our primary focus was to accelerate the training of single-spike SNNs using surrogate gradients and to tackle challenging temporal neuromorphic datasets (like the SHD dataset), which to date remains unexplored in single-spike SNNs. We did this because single-spike SNNs, when implemented on neuromorphic hardware, provide a very energy efficient alternative to multi-spike SNNs and standard ANNs. We have now made this more clear in the Introduction.
>
> **They use time-to-first-spike (TTFS) coding. After a neuron has fired, any eventual subsequent spike is blocked. Training is done with a surrogate gradient.**
>
> This is a common coding scheme used in many works and by no means a novelty of our work. Rather, we believe the novelty of our work to lie in the mathematical recasting of the slow sequential LIF operations with faster convolutional ones, which drastically speeds up training over other surrogate training methods (over 10x on some datasets)! Furthermore, we show for the first time how single-spike (i.e. TTFS) SNNs - with the inclusion of trainable membrane time constants - perform well on challenging neuromorphic datasets, reaching an accuracy of 70.32% on the SHD dataset (where an equivalent multi-spike SNN reaches an accuracy of 70.81%).
>
> **The authors limit themselves to extremely simple toy datasets (not even CIFAR!), and even on these datasets, the method is not competitive w.r.t. other proposals (see Table 1). Zhou et al AAAI 2021 should be included in Table 1. They have decent results on CIFAR, and even on ImageNet, using TTFS!**
>
> Thank you for pointing out the reference, we have updated our paper to include Zhou et al AAAI 2021 in Table 1. Their work is indeed impressive, yet the aim of our work and theirs is different. Zhou et al aim to (and succeed at) training very deep SNNs for image classification. In contrast, we aim to use the power of surrogate gradients in TTFS SNNs to tackle challenging neuromorphic datasets, where we developed a model which drastically accelerates training over the conventional approach, and which successfully obtains SOTA on the Yin-Yang and SHD datasets.
>
> Just as our model is not optimised for large scale image datasets, the model of Zhou et al is not optimised for challenging neuromorphic datasets, as their model uses IF neurons, where we show LIF neurons to be important for performance on challenging neuromorphic datasets. We agree that our explored image datasets are more toy-like. To emphasise, we included these experiments to demonstrate the versatility of our method to different problem domains and not with the aim of obtaining SOTA. However, the neuromorphic SHD dataset is by no means toy-like. For example, a multi-spike SNN with fixed membrane time constants only achieves an accuracy of 48.1% on this dataset ([Cramer et al. (2020)](https://ieeexplore.ieee.org/stamp/stamp.jsp?arnumber=9311226)), where we were able to obtain an accuracy of 70.32% using our single-spike model.
>
> Following your request - and to explore the applicability of our model to deeper networks - we have trained a 7 layer spiking convolutional VGG network using our method on the CIFAR10 dataset and obtained an accuracy of 82.14% (see [notebook](<https://github.com/webstorms/Block/blob/main/notebooks/review/VGG%20CIFAR10%20.ipynb>)). As expected, the performance is worse than that of Zho et al. (92.68%). However, they employed a deeper 16 layer VGG network, trained for 320 epochs (we trained for 140), and appear to use extensive training techniques such as batch and local response normalisation (which we did not). It is not unreasonable to assume that these additions would also improve our performance - alongside hyperparameter searches - which we leave to future investigations (due to time and resource constraints).

---

> > ### Author Response · Authors · 2022-11-18
> > **Response to Reviewer HU9D (Part 2)**
> >
> > **There is a new method for training TTFS-SNNs, but it's unclear what the advantages are w.r.t. existing methods.**
> >
> > The advantages of our method over other TTFS-SNNs are:
> > 1. Overcoming training problems: Unlike other TTFS-SNN methods, we use surrogate gradients which overcome the dead neuron problem (where lack of spike activity can otherwise halt learning in other TTFS-SNN methods). We show how our model is able to learn even when starting with zero spike activity (see Figure 4d).
> > 2. Training neural parameters other than synaptic weights. Most TTFS-SNNs train IF neurons, where we provide a training method that trains TTFS LIF neurons (including individual neuron time constants).
> > 3. Tackling challenging temporal datasets. Most TTFS-SNN are focused on non-temporal static datasets (like images). Using our new method, we demonstrate for the first time the applicability of using TTFS-SNNs on challenging temporal neuromorphic datasets, such as the SHD dataset (our model: 70.32% vs multi-spike SNN: 70.81%, whilst being 4x faster to train)!

---

> > > ### Comment · Reviewer_HU9D · 2022-11-21
> > > **Still unconvinced**
> > >
> > > Thank you for your responses.
> > >
> > > In my opinion, TTFS is not suitable for dynamical, continuous, inputs like SHD, because you cannot restrict the activity to at most one spike per neuron forever! You need to unblock spikes every now and then. I guess the authors do that between files. That's OK for offline processing, but how would you do that online?
> > >
> > > The accuracy they reach on CIFAR10 is well below the SOTA.
> > >
> > > Therefore I maintain my rating.

---

### Official Review · Reviewer_1hwt · 2022-10-28

**Confidence:** 4
**Correctness:** 3
**Technical Novelty And Significance:** 2
**Empirical Novelty And Significance:** 2
**Recommendation:** 5

**Clarity, Quality, Novelty And Reproducibility:**

Clarity is good.

Reproducibility should be good as the authors provided the code.

I did not see much novelty.


**Strength And Weaknesses:**

Strength:
+1. The submission is well written.

Weaknesses:
-1. Single spike SNN itself does not make sense to me, which is against the bio-plausibility of SNN. Essentially, the single-spike scheme dramatically impacts the effectiveness of SNN's temporal information. The authors also pointed it out in the section "Single-spike neurons solve challenging temporal problems using neural heterogeneity" and remediated the problem by learning time constants. But, the accuracy is much worse than Perez-Nieves'.

-2. The only motivation for using single-spike SNN is to enhance energy efficiency. But, the authors did not show any quantized experimental results to support the motivation. How much energy could it save on neuromorphic hardware?

-3. For LIF, we normally set the reset potential to 0. Is "calculate membrane potentials without reset" still valuable if that is the case?

-4. The motivation for training speedup is unclear to me. Yes, the 17x looks awesome. But how long does it take to train a multi-spike SNN model for the tested tasks?



**Summary Of The Paper:**

The authors proposed a new modal for training single-spike SNNs, which offers a 13.98x training speedup compared to a multi-spike counterpart. The speedup is achieved by avoiding all sequential dependence on time and exclusively relies on GPU parallelizable non-sequential operations. The effectiveness of the proposed method is reflected by five different datasets, performing on par with multi-spiking counterparts.

**Summary Of The Review:**

To me, the valuable part of SNNs is their bio-plausibility, especially in the temporal domain. Single-spike itself does not make sense to me, and the authors did not show the quantized experimental results on energy saving. Thus, the motivation for this work is unclear to me. In addition, the current neuromorphic hardware can barely mimic the simplest biological neuron. The fundamental research in the SNN domain should not be limited or favor current neuromorphic hardware. Instead, fundamental research should lead the development of neuromorphic hardware.

At the moment, I would not recommend this work. But I would like to see the authors' response.

---

> ### Author Response · Authors · 2022-11-18
> **Response to Reviewer 1hwt (Part 1)**
>
> Thank you for taking the time to review our submission. We hope our responses address your concerns and we welcome any further questions you may have.
>
> **Weaknesses: -1. Single spike SNN itself does not make sense to me, which is against the bio-plausibility of SNN.**
>
> We agree that the single-spike does not make much sense from a biological plausibility standpoint (although see [Heil, 2004](https://www.sciencedirect.com/science/article/pii/S0959438804000984); [Gollisch & Meister, 2008](https://www.science.org/doi/pdf/10.1126/science.1149639)). The value of single-spike networks is in their potential engineering applications in low-energy-cost computation, hence interest in them has recently been growing ([Comsa et al., 2020](https://ieeexplore.ieee.org/stamp/stamp.jsp?arnumber=9053856); [Oh et al., 2021](https://ieeexplore.ieee.org/stamp/stamp.jsp?arnumber=9439534); [Liang et al., 2021](https://ieeexplore.ieee.org/stamp/stamp.jsp?arnumber=9401607); [Eshraghian et al., 2021](https://arxiv.org/pdf/2109.12894.pdf); [Zenke et al., 2021](https://www.sciencedirect.com/science/article/pii/S089662732100009X)). Energy efficiency has been argued to be a key challenge in computing today ([Ionescu 2017](https://ieeexplore.ieee.org/stamp/stamp.jsp?arnumber=8268307)). Despite their potential value in energy efficiency, the challenges of single-spike SSNs are that their training is slow and difficult, and they can struggle with temporal stimuli. Hence, we set out to accelerate the training of these single-spike SNNs and enable them to work well with temporal stimuli.
>
> **Essentially, the single-spike scheme dramatically impacts the effectiveness of SNN's temporal information.**
>
> This is true, and hence a longstanding dogma has been that single-spike SNNs are not applicable for temporal problems. Notably - when including trainable membrane time constants - we demonstrate how our single-spike SNN (70.32%) is able to perform on a par with a multi-spike SNN (70.81%) on the challenging neuromorphic temporal SHD dataset (whilst our model trains 4x faster). Thus, we show that single-spike SNNs can be used to used tackle temporally challenging problems.
>
> **The authors also pointed it out in the section "Single-spike neurons solve challenging temporal problems using neural heterogeneity" and remediated the problem by learning time constants. But, the accuracy is much worse than Perez-Nieves'.**
>
> This drop in accuracy is expected (and would affect all single-spike SNNs, not just ours) as the Perez-Nieves model is multi-spike and employs recurrent connectivity. There is no doubt that multi-spike SNNs with recurrent connectivity are computationally superior to non-recurrent single-spike SNNs. We included the results [Perez-Nieves et al. (2021)](https://www.nature.com/articles/s41467-021-26022-3) to showcase the performance gap. However, single-spike networks are still of value due to their improved energy efficiency, and as we have demonstrated, rapid trainability.
>
> **2. The only motivation for using single-spike SNN is to enhance energy efficiency. But, the authors did not show any quantized experimental results to support the motivation. How much energy could it save on neuromorphic hardware?**
>
> The energy reduction % would be (approximately) the same as the spike reduction % (Figure4c), apologies we did not make this more clear. Using the same methodology as [Yin et al. (2020)](https://dl.acm.org/doi/pdf/10.1145/3407197.3407225) and [Panda et al. (2020)](https://www.frontiersin.org/articles/10.3389/fnins.2020.00653/full), the required energy for emulating a layer of LIF neurons on neuromorphic hardware is $E_{LIF}=(mn)E_{AC} F_r$, where m and n are the number of input and output neurons respecitvely, $F_r$ is the average firing rate, and $E_{AC}$ is energy cost for an accumulate operation (i.e. the energy required to store an input spike). All variables - between an architecturally identical single-spike and multi-spike SNN - are the same within the energy calculation formula, apart from the average firing rate. Thus, the % reduction in spikes implies an identical % reduction in energy consumption. To summarise the energy reduction between our single-spike and the multi-spike SNN:
> | Dataset | Energy reduction % |
> | --- | --- |
> | Yin-Yang | 44.22±1.32 |
> | MNIST | 49.75±2.48 |
> | F-MNIST | 52.75±0.70 |
> | N-MNIST | 81.35±0.43 |
> | SHD | 72.15±0.03 |

---

> > ### Author Response · Authors · 2022-11-18
> > **Response to Reviewer 1hwt (Part 2)**
> >
> > **3. For LIF, we normally set the reset potential to 0. Is "calculate membrane potentials without reset" still valuable if that is the case?**
> >
> > We only use the integrated membrane potentials $\tilde{V}$ without reset, as these are fast to compute and we can calculate the correct (first) spike time using them. If required, membrane potentials can be reset to 0 after spike by taking the hadamard product of the integrated membrane potentials $\tilde{V}$ with mask $m = z > 0$ (which is binary mask with $m[t]=1$ for every value $t>t1$, where t1 is the first spike time).  We have written you a [notebook](https://colab.research.google.com/drive/19ezOXIZeFq7KHD2UCYvJJ-jXVekTPbtQ?usp=sharing) to demonstrate how this can be done.
> >
> > **4. The motivation for training speedup is unclear to me.**
> >
> > Currently training SNNs is painfully slow. In training spiking networks, one also often needs to extensively explore hyperparameter space. Training speed ups allow for the space to be more efficiently explored, allowing more rapid research, development and construction of these networks. Furthermore, long training times imply longer GPU utilisation which is expensive (due to high energy bills) and environmentally unfriendly (due to the large energy consumption) ([Strubell et al. 2019](https://arxiv.org/pdf/1906.02243.pdf%22%3EWachstum), [Schwartz et al. 2020](https://dl.acm.org/doi/pdf/10.1145/3381831)).
> >
> > **Yes, the 17x looks awesome. But how long does it take to train a multi-spike SNN model for the tested tasks?**
> > Thank you! The training time and speedup of our single-spike model vs the multi-spike can be summarised as follows (all experiments were run on a A100 GPU):
> >
> > | Dataset | Multi-spike epoch train time (sec) | Our model epoch train time (sec) | Our model train speedup |
> > | --- | --- | --- | --- |
> > | Yin-Yang | 15.41±0.19 | 1.41±0.02 | 10.90x |
> > | MNIST | 62.19±0.14 | 14.06±0.01 | 4.42x |
> > | F-MNIST | 65.90±0.17 | 16.17±0.05 | 4.07x |
> > | N-MNIST | 181.59±0.05 | 41.93±0.35 | 4.33x |
> > | SHD | 45.81±0.19 | 11.28±0.17 | 4.06x |
> >
> > We have updated the Appendix (last page) of our paper to include these speedup results.
> >
> > **I did not see much novelty.**
> >
> > We believe the novelty of our paper to lie in the mathematical recasting of the otherwise slow sequential SNN training operations with faster convolutional ones, which dramatically accelerates training. Secondary to this, we illustrate how our model is able to reduce number of spikes over multi-spike SNNs, yet still obtain comparable accuracies. Notably - and for the first time - we show how single-spike SNNs perform on a par with multi-spike SNNs on challenging temporal datasets when using trainable membrane time constants; Thus suggesting - and conforming with your belief - that increasing biological realism plays an import role in the development of SNNs.
> >
> > **To me, the valuable part of SNNs is their bio-plausibility, especially in the temporal domain. Single-spike itself does not make sense to me, and the authors did not show the quantized experimental results on energy saving. Thus, the motivation for this work is unclear to me. In addition, the current neuromorphic hardware can barely mimic the simplest biological neuron. The fundamental research in the SNN domain should not be limited or favor current neuromorphic hardware. Instead, fundamental research should lead the development of neuromorphic hardware.**
> >
> > We hope we were able to address your questions and concerns. As computational neuroscientist we concur with the view that neuromorphic hardware development could benefit more from fundamental neuroscience research. However, we think there is also space to consider SNN variants, such as the single-spike SSN, that may be valuable for their engineering applications rather than their bio-plausibility. We hope that our new explanations and analyses now demonstrate the potential energy efficiency advantages of single-spike SNNs. We would also add that we are currently developing faster-to-train versions of standard multi-spike SSNs based on our developments here with single-spike SNNs.
> >
> > **At the moment, I would not recommend this work. But I would like to see the authors' response.**
> >
> > We thank you for being open minded and we hope we were able to address your concerns and questions.

---

> > > ### Comment · Reviewer_1hwt · 2022-11-21
> > > **Thanks for the feedback!**
> > >
> > > I appreciate the authors' efforts in answering all reviewers' questions! The authors addressed all my questions!
> > >
> > > However, I am still not convinced by the "single-spike" scheme, which eliminates the temporal cues from SNNs. My personal thoughts on SNNs are that the community should focus on the bioplausible side and lead the development of neuromorphic hardware at the moment. Suppose we lower the weights on the bioplausible part. In that case, the motivation for studying SNNs is less meaningful, especially since ANN-based approaches offer better performances on the tasks used in this submission.
> > >
> > > In addition, I would suggest showing the energy efficiency based on real neuromorphic hardware instead of simulation.
> > >
> > > In summary, I keep my original score after reviewing the rebuttal and other reviewers' comments.

---

### Decision · Program_Chairs · 2023-01-20

**Decision:**

Reject

**Justification For Why Not Higher Score:**

This paper should be rejected for sure.

**Justification For Why Not Lower Score:**

N/A

**Metareview: Summary, Strengths And Weaknesses:**

The paper got two 5s (marginally above threshold) and two 3s (reject). The major challenges include unconvincing motivation (especially the "single-spike" scheme), limited novelty, weak experiments, etc. The author rebuttals did not address all the reviewers' concerns and all the reviewers kept their scores. By the overall scores, the AC recommended rejection.

**Summary Of Ac-Reviewer Meeting:**

N/A